# Epithelial cells release adenosine to promote local TNF production in response to polarity disruption

Ingrid Poernbacher[1] & Jean-Paul Vincent[1]

Disruption of epithelial integrity contributes to chronic inflammatory disorders through persistent activation of stress signalling. Here we uncover a mechanism whereby disruption of apico-basal polarity promotes stress signalling. We show that depletion of Scribbled (Scrib), a baso-lateral determinant, causes epithelial cells to release adenosine through equilibrative channels into the extracellular space. Autocrine activation of the adenosine receptor leads to transcriptional upregulation of TNF, which in turn boosts the activity of JNK signalling. Thus, disruption of cell polarity feeds into a well-established stress pathway through the intermediary of an adenosine signalling branch. Although this regulatory input could help ensuring an effective response to acute polarity stress, we suggest that it becomes deleterious in situations of low-grade chronic disruption by provoking a private inflammatory-like TNF-driven response within the polarity-deficient epithelium.

[1] The Francis Crick Institute, Midland Rd 1, London NW1 1AT, UK. Correspondence and requests for materials should be addressed to I.P. (email: Ingrid.Poernbacher@crick.ac.uk)

Acute disruption of epithelial integrity is usually rapidly repaired through a transient response involving Jun N-terminal kinase (JNK) signalling[1]. However, prolonged activation of JNK-mediated stress is deleterious and leads to inflammation[2,3]. Indeed, JNK signalling is a known contributor to pathologies associated with chronic epithelial damage (fibrosis or chronic infection) and a driver of malignant progression[4–6]. Therefore, understanding the mechanisms that trigger and sustain JNK signalling in response to epithelial stress is of utmost interest. A common feature of epithelial stress is the disruption of cell polarity. Indeed, removal of the baso-lateral determinant Scrib leads to activation of JNK signalling in *Drosophila*[7–10] and mammalian systems[11,12]. Genetic analysis of this process has been hindered by the confounding effect of cell competition, whereby defective cells are eliminated when confronted with normal cells[13]. For example, in *Drosophila* wing precursors small groups of *scrib* mutant cells activate JNK—in a manner that requires TNF$^{Egr}$ (a Tumor Necrosis Factor) and its cognate receptor (TNFR$^{Grnd}$)—before being eliminated by apoptosis[7–10,14] (Supplementary Fig. 1a). Upon blockade of cell competition by inhibition of PTP10D signalling, these small groups of *scrib* mutant cells maintain JNK signalling but are not eliminated and contribute adult tissue, albeit in a highly abnormal fashion[10]. Therefore, in the absence of cell competition, *scrib* mutant cells survive but remain subject to persistent stress signalling, which prevents them from contributing normal tissue[10]. Another means of circumventing the confounding effects of cell competition is to disrupt cell polarity in a broad domain, thus preventing defective cells from interacting with normal cells. Indeed, no excess apoptosis takes place in imaginal discs wholly lacking Lgl, another baso-lateral determinant[15]. Likewise, only a few dying cells were seen within a broad band of cells expressing an RNAi transgene against *scrib* under the control of *spalt(sal)*-Gal4 (see below). Here, we show that such sub-apoptotic chronic perturbation of polarity prompts epithelial cells to release adenosine in the extracellular space. Adenosine then acts through the adenosine receptor (AdoR) to activate the local production of TNF and subsequent JNK signalling.

## Results

**AdoR boosts JNK signalling during chronic polarity stress.** In *sal> scrib-Ri* discs, JNK signalling was activated throughout the domain of polarity disruption, as indicated by the up-regulation of a transcriptional reporter comprising multimerised AP1 binding sites driving the expression of dsRed (*TRE-dsRed*[16]) (Fig. 1a, b, d). We conclude that disruption of cell polarity elicits non-apoptotic persistent stress signalling. To confirm the involvement of JNK in response to polarity disruption, we co-expressed Puckered (Puc), an inhibitory phosphatase, along with the RNAi against *scrib*. The activity of *TRE-dsRed* returned to the background level (Fig. 1c, d). Likewise, expression of a dominant negative form of JNK (here referred to as *JNK$^{bskDN}$*) prevented the upregulation of *TRE-dsRed* in *scrib*-deficient cells (Fig. 1d). We conclude that the response to large-scale disruption of cell polarity requires core JNK signalling. Importantly, such persistent JNK signalling is deleterious since the resulting wings were severely damaged, a phenotype that was partially alleviated by suppression of JNK signalling (Fig. 1e–g, Supplementary Fig. 1b). To test the requirement of TNF$^{Egr}$ and TNFR$^{Grnd}$, known activators of JNK signalling, we expressed highly effective RNAi transgenes against these components (see Methods for RNAi validation), along with *scrib*-RNAi. RNAi against *TNFR$^{grnd}$* fully suppressed the JNK activity triggered by *scrib* knockdown (Fig. 1h, k). This result suggests that polarity disruption autonomously leads to JNK signalling through activation of the

TNFR$^{Grnd}$ receptor. By contrast, knocking down *TNF$^{egr}$* in a similar experimental set up (with an RNAi line that completely suppresses the effect of overexpressed TNF$^{Egr}$) only resulted in partial rescue (Fig. 1i, k), suggesting that some, but not all ligand originates from the disrupted tissue. Removal of *TNF$^{egr}$* in the whole animal (in a null mutant background) was fully effective at suppressing JNK signalling (Fig. 1j, k). We conclude that the canonical JNK signal transduction pathway is autonomously required for the stress response to polarity disruption and that the ligand (TNF$^{Egr}$) originates both from within and without the polarity-deficient tissue.

We next set out to uncover the mechanism that senses polarity disruption and transmits the information to JNK signalling. While assessing the possible involvement of known stress-related pathways, we found, unexpectedly, that knocking down the ATP synthase and other components of mitochondrial electron transport suppressed JNK signalling in *sal> scrib-Ri* discs (Supplementary Fig. 1c). Such suppression was surprising because these genetic manipulations are expected to exacerbate cellular stress by increasing ROS and reducing intracellular ATP levels[17]. Beside their well-known role in energy metabolism, mitochondria have been shown to affect the availability of extracellular ATP, which can act as a signalling molecule[18]. However, no recognizable receptor of extracellular ATP has been identified in *Drosophila*. Nevertheless, ATP can be metabolized into adenosine, for which a receptor has been identified: the GPCR encoded by the *AdoR* gene[19–24]. *Drosophila AdoR* mutants are viable and show no overt phenotype[20,25]. To test the possible role of adenosine signalling in response to chronic polarity disruption, we assessed the effect of *AdoR* knockdown on JNK activity in *scrib*-deficient tissue. As shown in Fig. 1l, n, this led to strong (though not complete) suppression of *TRE-dsRed* activity, as well as partial rescue of wing defects (Fig. 1o, Supplementary Fig. 1b). A similar result was found in an *AdoR* mutant background (Fig. 1m, n, p, Supplementary Fig. 1b). Loss of *AdoR* (by knockdown or mutation) also suppressed excess JNK activity caused by expression of scrib-RNAi with the strong *engrailed* (*en*)-Gal4 driver (*en> scrib-Ri*) (Supplementary Fig. 1d–f). The role of AdoR was further tested in wing imaginal discs obtained from homozygous *scrib* mutant larvae (Supplementary Fig. 1g–k). These discs, which undergo tumorous growth[26], upregulated the JNK signalling sensor (Supplementary Fig. 1g, h), as well as the established JNK target genes *puc* and *dilp8* (Supplementary Fig. 1j). Expression of these genes was brought back toward normal levels by concomitant removal of *TNF$^{egr}$* or its receptor *TNFR$^{grnd}$*, confirming the involvement of this ligand/receptor pair (Supplementary Fig. 1j). The same effect was seen with the removal of *AdoR* (Supplementary Fig. 1i, j), an indication that AdoR contributes to JNK activation in these tumors, as it does in *sal> scrib-Ri* discs. Consistent with the established role of JNK signalling in tumor growth, removal of *AdoR* (or *TNFR$^{grnd}$*) also led to a reduction in tumor size (Supplementary Fig. 1k), suggesting a possible role for adenosine signalling in epithelial tumor growth. Of note, inhibition of adenosine signalling did not prevent the elimination of small patches of *scrib* mutant cells by cell competition: *scrib AdoR* double mutant clones were outcompeted as efficiently as *scrib* single mutant clones (Supplementary Fig. 1l, m), even though the removal of *AdoR* from *scrib* mutant clones caused a small, albeit significant, reduction in JNK signalling (Supplementary Fig. 1n). We conclude that activation of JNK signalling in *scrib* mutant cells during cell competition is largely, but not completely, AdoR-independent, while AdoR signalling is a major contributor to JNK-signalling during chronic epithelial polarity disruption.

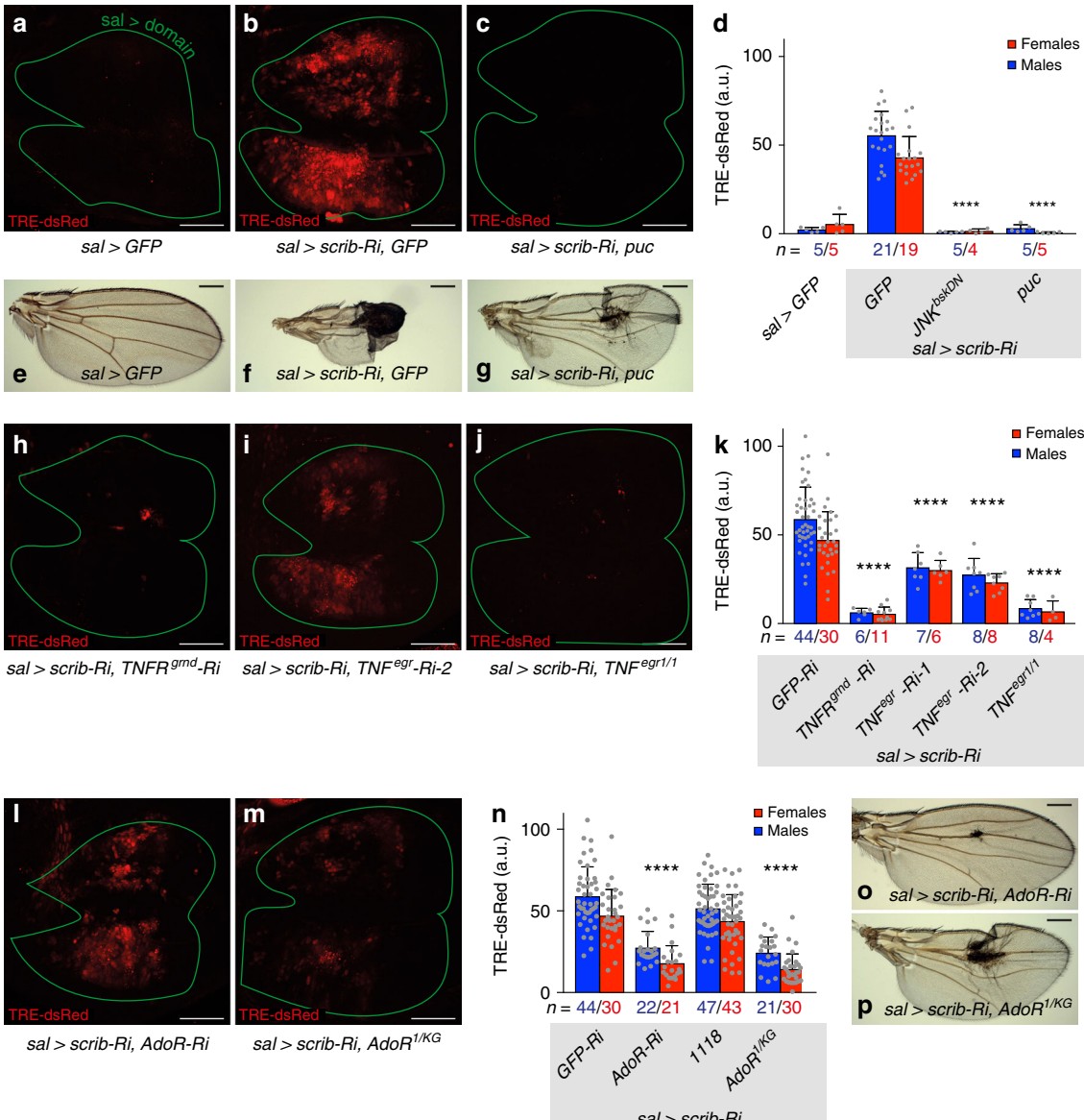

**Fig. 1** AdoR signalling enhances JNK pathway activity during chronic polarity stress. **a–g** Silencing *scrib* (*sal> scrib-Ri*) activates the JNK sensor *TRE-dsRed* in the wing disc and causes severe damage in the adult wing. These effects are suppressed by expression of *JNK^{bskDN}* or the inhibitor of JNK, *puc*.
**h–k** Expression of *TRE-dsRed* is also suppressed by knockdown of *TNFR^{grnd}* or in a *TNF^{egr}* mutant. It is partially suppressed by *TNF^{egr}* knockdown.
**l–p** Similarly, *TRE-dsRed* is suppressed and wing damage is rescued following *AdoR* knockdown or in an *AdoR* mutant. Scale bars, 50 μm (**a–c**, **h–j**, **l**, **m**) and 0.5 mm (**e–g**, **o**, **p**). In graphs, means are shown, and error bars represent ± SD; ****$P < 0.0001$, unpaired two-tailed Student's *t*-test (males and females pooled together). As in all subsequent figures, confocal images are maximal intensity projections of *z*-stacks, male third instar wing imaginal discs are shown with posterior to the right and dorsal up and a green line marks the domain of *sal*-Gal4 expression, as determined by the activation of a *UAS-cd8GFP* transgene (see detailed genotypes in Methods). **e–g**, **o**, **p** are representative images of male adult wings

**Polarity-deficient cells release adenosine to activate AdoR.** Our data suggest that loss of polarity could cause epithelial cells to release adenosine, a known danger signal[27–32], which would then activate AdoR and boost JNK signalling. Levels of extracellular adenosine (e-Ado) are low under healthy conditions[24] but are expected to rise upon overexpression of an equilibrative adenosine transporter by allowing the efflux of intracellular adenosine (Supplementary Fig. 2a). Overexpression of one such equilibrative adenosine transporter Ent2 (in *sal> ent2* imaginal discs) led to an increase in JNK signalling (Fig. 2a, b, d), which was suppressed by co-expression of *AdoR*-RNAi (Fig. 2c, d) and markedly enhanced by co-overexpression of AdoR (Fig. 2f, g). These results confirm our suggestion that e-Ado boosts JNK signalling in an AdoR-dependent fashion. As a further test, we assessed the effect of

incubating imaginal discs ex-vivo with the stable adenosine receptor agonist 2-chloroadenosine (CADO). Wild-type imaginal discs responded by increased expression of the JNK target gene *dilp8* (Fig. 2h). No such increase was seen in similarly treated imaginal discs explanted from *AdoR* mutants, while discs over-expressing AdoR showed even higher *dilp8* expression (Fig. 2h). Together the above results suggest that e-Ado triggers or boosts JNK signalling through AdoR. The spatial range of e-Ado is probably limited since JNK signalling is cell-autonomously activated in Ent2 overexpressing clones (Supplementary Fig. 2b–e). Such short-range action confirms earlier suggestions that e–Ado has a short half-life in the extracellular space[33].

Overexpression of AdoR alone in otherwise wild-type tissue has only a minor effect on JNK signalling (Fig. 2e), suggesting

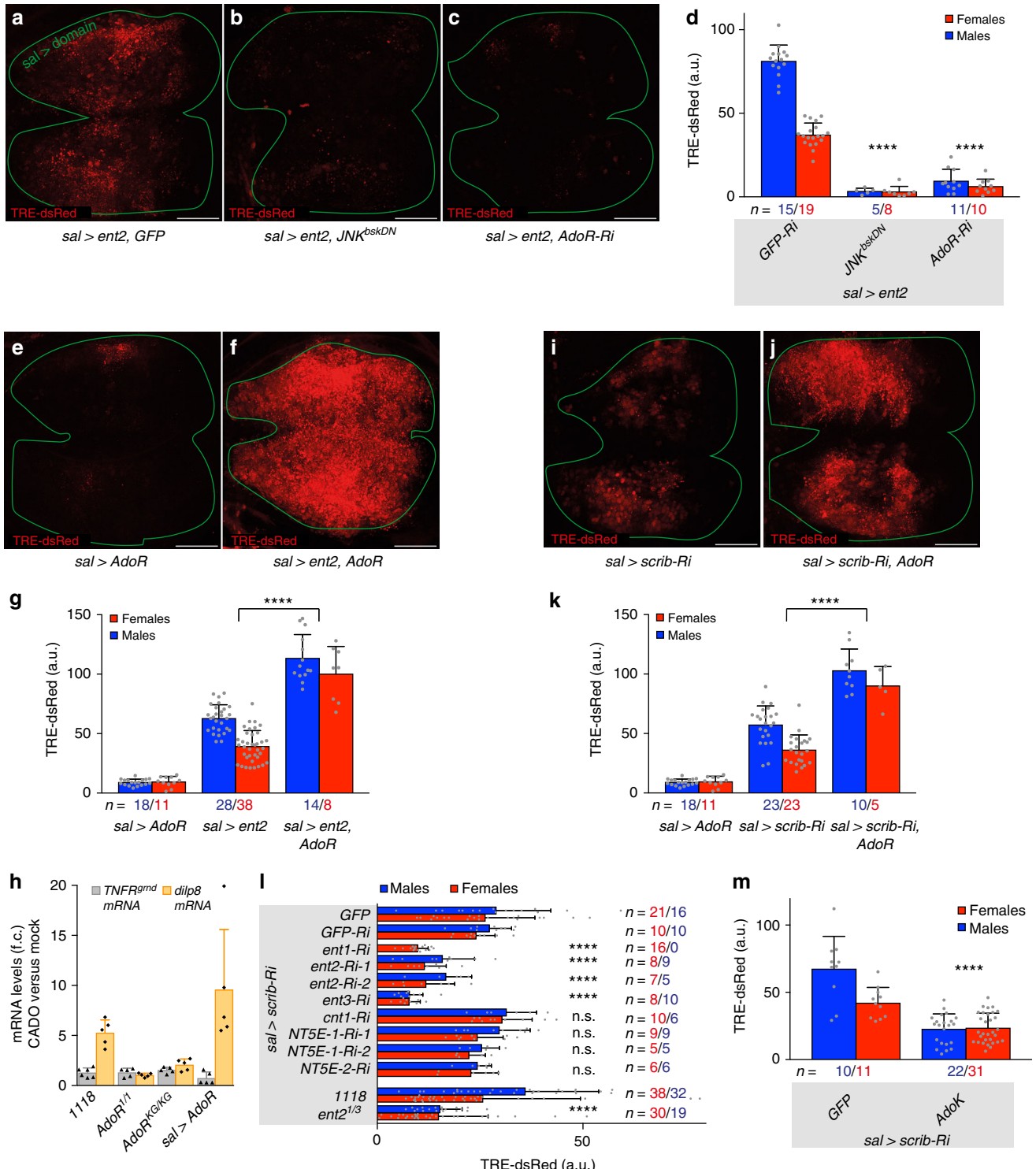

**Fig. 2** AdoR is activated by adenosine released from polarity-deficient epithelial cells. **a**–**d** Ent2 overexpression (*sal> ent2*) activates *TRE-dsRed* in the wing disc. This is suppressed by expression of *JNK^bskDN* or silencing of *AdoR*. **e**–**g** AdoR overexpression (*sal> AdoR*) causes a very mild activation of *TRE-dsRed* on its own, but strongly enhances JNK activation caused by overexpression of Ent2. **h** CADO treatment increases expression of the JNK target gene *dilp8* in explanted wild-type, but not in *AdoR* mutant discs. This effect is further enhanced by AdoR overexpression. Here, *TNFR^grnd* expression, measured by RT-qPCR, is shown as a control. Fold changes are relative to *rp49*, $n \geq 5$. **i**–**k** AdoR overexpression leads to a strong enhancement of *scrib*-RNAi mediated activation of JNK signalling. **l**, **m** Silencing of various *equilibrative adenosine transporter genes* (*ent1*, *ent2*, *ent3*) by RNAi, reduction of Ent2 activity in a hypomorphic background, *ent2^{1/3}* (**l**) or overexpression of AdoK (**m**) suppress *scrib*-RNAi mediated activation of JNK, whereas silencing of *concentrative adenosine transporter* (*cnt1*), *ecto-nucleotidase NT5E-1* or *NT5E-2* had no effect (**l**). Experimental RNAi lines in (**l**) were compared to 'GFP-Ri' for statistical analysis. Scale bars, 50 μm. In graphs, means are shown, and error bars represent ±SD; ****$P < 0.0001$, ^{n.s.}$P \geq 0.05$, unpaired two-tailed Student's *t*-test (males and females pooled together)

that, under normal circumstances, little adenosine is present in the extracellular space. However, AdoR overexpression strongly enhanced the pro-JNK effect of *scrib* knockdown (Fig. 2i–k). One likely interpretation is that *scrib*-deficient cells release adenosine, whose effect is magnified by the excess AdoR. How could polarity-deficient cells release adenosine? One trivial possibility is that these cells are on the brink of apoptosis and, as part of this process, would release ATP, which would then be converted into adenosine in the extracellular space. This seems unlikely as only a small number of apoptotic cells can be seen in *scrib* deficient tissue (Supplementary Fig. 2f). Indeed, overexpression of p35, an inhibitor of apoptosis, did not reduce JNK signalling in *scrib*-deficient cells (Supplementary Fig. 2g). Moreover, it is unlikely that e-Ado originates from the conversion of extracellular ATP because preventing such conversion by knocking down the *ecto-nucleotidases* that metabolize extracellular ATP into adenosine (*NT5E-1* and *NT5E-2*) did not bring down JNK signalling in *scrib*-deficient cells (Fig. 2l). In contrast, overexpression of adenosine kinase (AdoK), which converts intracellular adenosine to AMP and is hence expected to reduce the level of intracellular adenosine, did reduce JNK signalling (Fig. 2m). Therefore, adenosine itself could be released from polarity-challenged cells, either through active nucleoside transporters or passive nucleo-side channels[28,30]. Knockdown of a *concentrative nucleoside transporter* (*cnt1*) did not bring down JNK signalling triggered by *scrib*-RNAi (Fig. 2l), whereas knockdown of *equilibrative nucleo-side transporters* (*ent1, ent2* and *ent3*) or a mutation in *ent2* did (Fig. 2l). This was confirmed at the level of the terminal phenotype since depletion of *ent2* led to partial rescue of wing defects in *sal> scrib-Ri* animals (Supplementary Fig. 2h–k). How could polarity disruption cause adenosine efflux? It is unlikely to be mediated by a change in the subcellular localization or levels of the transporters since, at least for Ent2, these were unaffected by *scrib* knockdown (as seen with Ent2-GFP expressed from a genomic fosmid) (Supplementary Fig. 2m). One possible scenario is that mechanical strain caused by alterations of epithelial architecture could open up equilibrative nucleoside transporters. Indeed, such a mechanism has been proposed to release ATP through pannexin channels[34]. We cannot, however, exclude the alternative possibility that *scrib*-deficient cells increase the production of adenosine, stimulating efflux into the extracellular space. Irrespective of the mechanism, our results imply that polarity stress leads to adenosine efflux through equilibrative nucleoside transporters.

**AdoR acts upstream of TNF[Egr].** Next, we sought to find out how AdoR signalling feeds onto JNK signalling. *Drosophila* AdoR has been shown to signal through the G protein/cAMP-dependent Protein Kinase A (PKA) pathway[20,23]. Consistently, expression of $G\alpha_s$-*RNAi*, dominant negative PKA (*PKAmR\**), or *PKA-C-RNAi* led to a reduction of *TRE-dsRed* expression in *sal> scrib-Ri* discs (Supplementary Fig. 3a–d). Therefore, the effect of AdoR on JNK signalling in polarity deficient discs is probably mediated by $G\alpha_s$/cAMP/PKA. We then devised epistasis tests to determine where AdoR signalling impinges on JNK signal transduction. AdoR signalling was activated by co-expressing Ent2 and AdoR or *scrib*-RNAi and AdoR. In both cases, the resulting upregulation of *TRE-dsRed* was suppressed by RNAi against *TNFR[grnd]*, confirming that TNFR[grnd] and downstream JNK signalling are activated in response to AdoR signalling (Fig. 3a, b). *TRE-dsRed* expression was also suppressed by RNAi against *TNF[egr]* or in a *TNF[egr]* mutant background (Fig. 3a, b). Overexpression of AdoR alone triggered mild activation of *TRE-dsRed* expression and this too was suppressed by removal of *TNF[egr]* (as well as *TNFR[grnd]*) (Fig. 3c–g). Therefore, the complete JNK signal transduction

pathway, from the TNF[Egr] ligand down to JNK itself is needed for AdoR signalling to stimulate JNK signalling. Conversely however, JNK activation induced by overexpressed TNF[Egr] was not affected by *AdoR*-RNAi (Fig. 3h–j), showing that AdoR is not a core component of TNF/JNK signalling. One possibility is that AdoR signalling stimulates local TNF[Egr] expression.

**AdoR acts as a transcriptional regulator of TNF[Egr].** To assess the effect of AdoR signalling on TNF[Egr] expression, we measured the levels of *TNF[egr]* transcripts in imaginal discs overexpressing Ent2 and AdoR under the control of *sal*-Gal4. As predicted, they were markedly increased relative to the situation in control discs (Fig. 4a). This rise was unaffected by over-expression of Puc and is therefore independent of JNK signalling activity (Fig. 4a). As an additional assay for TNF[Egr] expression, we used a GFP-tagged TNF[Egr] expressed from a genomic fosmid[35]. We found that the upregulation of this transgene seen in *sal> ent2 AdoR* imaginal discs (Fig. 4b, c) was not diminished by inhibition of JNK with a *JNK[bskDN]* (Fig. 4d). In this situation, accumulation of TNF[Egr] protein is unlikely to originate from immune cells because none appear to be recruited to the *sal> ent2 AdoR* expressing domain (Fig. 4c; see more detail in Supplementary Fig. 4a, b). Overall, our analysis shows that the upregulation of TNF[Egr] by AdoR signalling results from a local transcriptional response. Consistent with this suggestion, we found increased *TNF[egr]* mRNA levels in wild-type discs treated with the adenosine receptor agonist CADO (Fig. 4e). This effect was enhanced in AdoR over-expressing discs and was prevented in *AdoR* mutant discs, underscoring the involvement of the adenosine receptor (Fig. 4e). Therefore, AdoR-mediated activation of *TNF[egr]* transcription could account for the activation of JNK signalling in *scrib*-deficient discs. Indeed, the level of *TNF[egr]* transcript was upregulated in *en> scrib Ri* discs (Fig. 4f) and this was suppressed in an *AdoR* mutant background or by concomitant expression of RNAi against *AdoR* or *ent2*, but not by over-expression of Puc (Fig. 4f). Unfortunately, we could not directly test whether the PKA pathway is required for increased *TNF[egr]* transcription in *en> scrib Ri* discs, because *PKAmR\** or *PKA-C-RNAi* under control of this driver led to early lethality. Consistent with the above findings in *Drosophila*, we found that *TNF-α* transcript levels were upregulated in adenosine-treated immortal human keratinocytes (HaCaT) (Fig. 4g), an effect that was inhibited by three different adenosine receptor antagonists (CGS15943, SCH58261 and Caffeine) or by a PKA inhibitor cocktail (Merck # 20-114), but not by a potent JNK inhibitor (SP600125) (Fig. 4h). Activation of TNF expression by extracellular adenosine could therefore be a general feature of epithelial cells.

Taken together, our results show that polarity disruption induces AdoR signalling, which promotes transcription of *TNF* and boosts JNK signalling. This mechanism could contribute to tissue repair following acute epithelial disruption. However, it could become deleterious in chronic low-grade situations. Indeed, suppression of AdoR signalling improves the phenotype of wings emerging from polarity-deficient imaginal discs (Fig. 1o, p). Therefore, at least in this situation, JNK signalling sustained by AdoR signalling contributes to tissue damage. As an additional test of this possibility, we assessed the survival of adult flies continuously fed with a low dose of Bleomycin, which damages the epithelial layer of the midgut, thus causing chronic intestinal stress[36]. We found that, under such conditions, *AdoR* mutants had a slightly improved lifespan compared to control wild-type animals (Fig. 4i). By contrast, *AdoR* mutants were significantly more sensitive to an acute stress (dry starvation) than wild-type animals[37] (Fig. 4j). This suggests that AdoR-mediated induction of TNF expression helps mount an effective response to acute

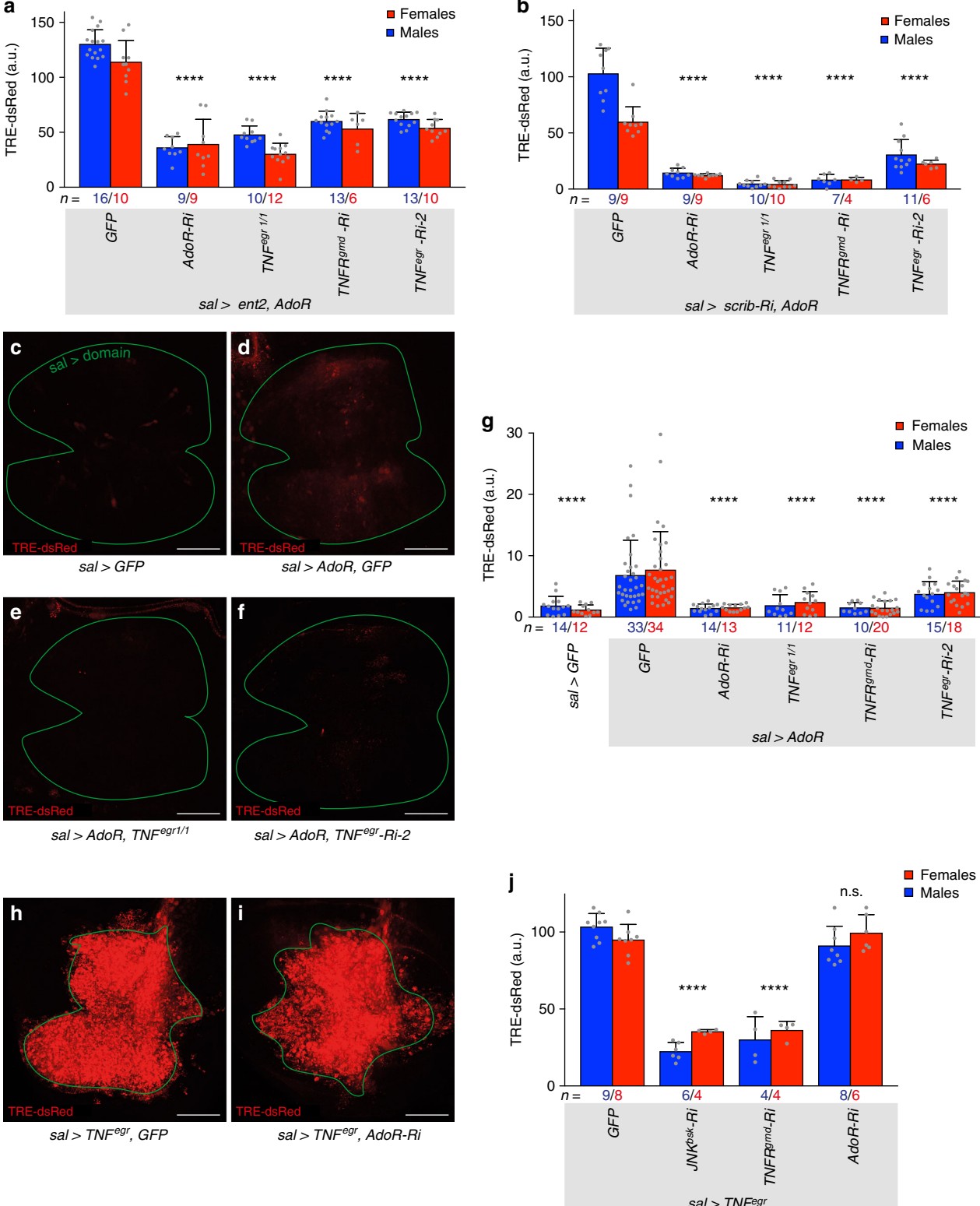

**Fig. 3** AdoR acts upstream of TNF^Egr during chronic polarity stress. **a**, **b** Activation of *TRE-dsRed* in *sal> ent2, AdoR* or in *sal> scrib-Ri, AdoR* discs is suppressed by expression of *AdoR*-RNAi, mutation in *TNF^egr*, expression of *TNFR^grnd*-RNAi or expression of *TNF^egr*-RNAi. **c–g** Mild activation of *TRE-dsRed* caused by overexpression of AdoR alone is rescued by *AdoR*-RNAi, in a *TNF^egr* mutant background, by *TNFR^grnd*-RNAi or by *TNF^egr*-RNAi. **h–j** *TRE-dsRed* activation triggered by overexpression of TNF^Egr is suppressed by *JNK^bsk*-RNAi or *TNFR^grnd*-RNAi but is not affected by *AdoR*-RNAi. This experiment was carried out at 29 °C. Scale bars, 50 μm. In graphs, means are shown, and error bars represent ±SD; ****$P < 0.0001$, ^n.s.$P \geq 0.05$, unpaired two-tailed Student's *t*-test (males and females pooled together)

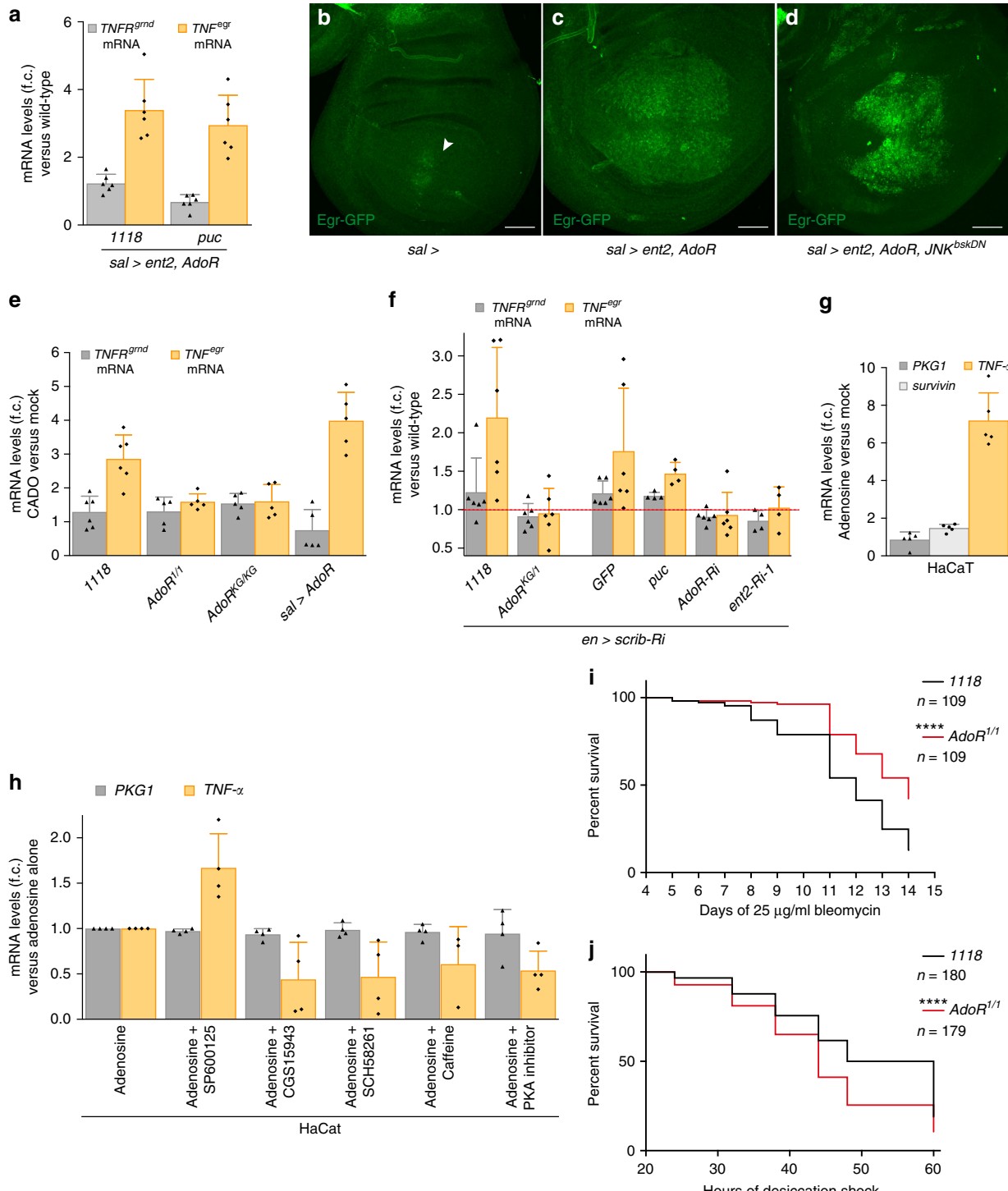

**Fig. 4** AdoR promotes transcription of TNF[Egr] in polarity-deficient epithelial cells. **a** *sal> ent2, AdoR* expression enhances *TNF[egr]* transcript levels in the wing disc independently of JNK signalling. *TNFR[grnd]* expression, measured by RT-qPCR, is shown as a control. Fold changes are relative to *rp49*, $n = 6$. **b–d** *sal> ent2, AdoR* expression upregulates expression of Egr-GFP from a genomic fosmid (**b**, **c**). This effect is not rescued by co-expression of *JNK[bskDN]* (**d**). The arrowhead in **b** indicates a suspected position effect of the attP site (VK00033) where the fosmid was integrated. **e** CADO treatment increases expression of *TNF[egr]* in explanted wild-type discs, but not in *AdoR* mutant discs. This effect is further enhanced by Ador overexpression. *TNFR[grnd]* expression, measured by RT-qPCR, is shown as a control. Fold changes are relative to *rp49*, $n \geq 5$. **f** *en> scrib Ri* discs upregulate *TNF[egr]* mRNA in an AdoR/Ent2-dependent and JNK-independent fashion. Fold changes are relative to *rp49*, $n \geq 4$. **g** Human HaCaT cells treated with adenosine upregulates *TNF-α* transcript levels (as assayed by qRT-PCR). Fold changes are relative to *GAPDH*, $n = 5$. Expression of *PKG1*, a housekeeping gene and *survivin*, a JNK target gene, are also shown. **h** The effect shown in **g** was suppressed by three different adenosine receptor antagonists (CGS15943, SCH58261 or Caffeine) and by a PKA inhibitor cocktail (Merck # 20-114), but not by JNK inhibitor SP600125. Fold changes are relative to *GAPDH*, $n \geq 3$ and expression of *PKG1* is shown. **i, j** *AdoR* mutants live longer than wild-type flies when continuously fed with a low dose of Bleomycin (**i**) but live shorter than wild-type flies during acute desiccation (dry starvation, **j**). ****$P < 0.0001$, Gehan–Breslow–Wilcoxon test. Scale bars, 50 µm. In graphs, means are shown, and error bars represent ±SD

**a**

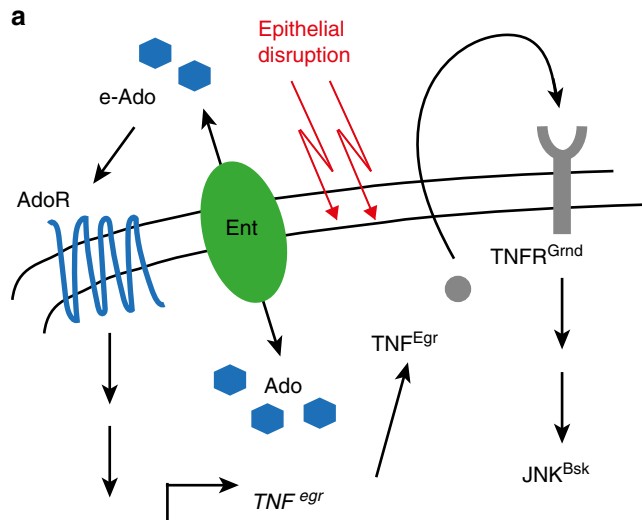

**Fig. 5** Summary model of AdoR signalling in polarity deficient discs. Polarity stress triggers the release of adenosine in the extracellular space. Subsequent activation of AdoR signalling leads to the production of TNF, which in turn activates JNK

insults while persistent JNK signalling sustained by AdoR signalling in response to chronic stress could be harmful.

## Discussion

We have shown that adenosine acts as a warning signal in response to sub-apoptotic perturbation of polarity in epithelial cells. Adenosine released in the extracellular space triggers the local production of TNF, which in turn activates JNK, a well-established stress mediator (Fig. 5). Because e-Ado is short-lived, it is unlikely to act at a long range. In fact, we suggest that a large number of contiguous cells need to be disrupted for them to collectively release adenosine at a level that is sufficient to elicit a response. This community effect would ensure that a stress response is only mounted when a significant amount of tissue is affected, as would occur in a situation of low-grade but widespread stress. In our model of perturbation of cell polarity, adenosine promotes TNF production within the affected epithelium itself, eliciting a private inflammatory-like response without the involvement of immune cells. Such a local, e-Ado-stimulated, response could be an important contributor to pathologies associated with chronic epithelial damage. The above scenario is in contrast with the orthodox view that it is the activated monocytes/macrophages that normally produce TNF and that this is inhibited by adenosine[38,39]. We speculate that the type of response elicited by tissue disruption depends on the severity of insult or type of pathology, a consideration that must be kept in mind while designing therapeutic strategies.

## Methods

**Drosophila stocks and genetics**. To generate UAS-AdoR, a genomic AdoR fragment was PCR amplified from BAC CH321-68K21 using the primers fwd 5′-CCACCATGTCCGCGTTTC-3′ and rev 5′-GATCCCGCTTCTTTTCCGAC-3′, cloned into p10xUASt-attB and integrated into attP site VK02 28E7. To generate UAS-AdoK, the AdoK coding sequence was PCR amplified from BDGP clone GH14845 using primers fwd 5′-ATGACGAGCACGCTACAAGAAGG-3′ and rev 5′-TTACTCAACGAACTCCGGTTCGC-3′, cloned into p10xUASt-attB and integrated into attP site ZH-86Fb 86F8.

UAS-ent2-HA (F002577) was provided by FlyORF. Ent2-GFP (Ent2-2XTY1-SGFP-V5-preTEV-BLRP-3XFLAG, ID318249) and Egr-GFP (Egr-2XTY1-SGFP-V5-preTEV-BLRP-3XFLAG, ID 318615) were provided by the VDRC Tagged FlyFos TransgeneOme Library (fTRG).

AdoR$^{KG}$ (BL30868), AdoR-GFP (AdoR$^{MI01202-GFSTF.1}$, BL60165), UAS-PKAmR* (BL35550), UAS-bskDN (BL6409), UAS-GFP (in VALIUM10 attP2, BL35786) and UAS-p35 (BL5072) were provided by the Bloomington Drosophila Stock Center.

white$^{1118}$ was provided by A. Gould. AdoR$^1$, ent2$^1$ and ent2$^3$ were provided by G. Boulianne[40]. TRE-dsRed (attP16, 2nd chromosome) and TRE-GFP (attP16, 2nd chromosome) were provided by D. Bohmann[16]; TRE-dsRed (attP VK00033, 3rd chromosome) was generated using the same plasmid as the Bohmann lab. Grnd$^{Minos}$ (Mi$^{14}$CG10176$^{MI05292}$) was provided by P. Leopold[14]. Additional fly strains were: UAS-puc$^{2A41}$, egr$^{142}$, UAS-egr (2nd chromosome$^{43}$) and scrib$^1$ $^{44}$.

The following RNAi lines were from the TRiP (Transgenic RNAi project) collection at Harvard Medical School: egr RNAi-1 (BL55276), AdoR RNAi (BL27536), Gα$_S$ RNAi (BL29576), PKA-C1 RNAi (BL31599), PKA-C2 RNAi (BL31656), PKA-R1 RNAi (BL27308), PKA-R2 RNAi (BL34983), NT5E-1 RNAi-2 (BL55195), COX5A RNAi (BL27548) and ATPsyn-α RNAi (BL28059). The following RNAi lines were from the GD collection of the Vienna Drosophila RNAi Center (VDRC): bsk RNAi (GD34139), grnd RNAi (GD43454), egr RNAi-2 (GD45253), ent1 RNAi (GD49328), ent2 RNAi-1 (GD7618), ent3 RNAi (GD47536), cnt1 RNAi (GD7374), NT5E-1 RNAi-1 (GD49359) and NT5E-2 RNAi (GD10051). The following RNAi lines were from NIG-FLY (National Institutes of Genetics, Japan): GFP RNAi (GFP-IR-1), scrib RNAi (5462R-2), ent2 RNAi-2 (31911R-3) and mRpL4 (5818R-1).

Validation of RNAi lines: AdoR RNAi (BL27536) rescues early larval lethality caused by overexpression of AdoR (act> AdoR) to pharate adults. grnd RNAi (GD43454) reduces ectopic TRE-dsRed expression in sal> grnd$^{intra}$ by ~90%. egr RNAi-1 (BL55276) and egr RNAi-2 (GD45253) reduce TRE-dsRed expression in sal> egr by ~100% and ~80%, respectively. bsk RNAi (GD34139) reduces the expression of TRE-dsRed seen in sal> egr by ~70%. ent2 RNAi-1 (GD7618) and ent2 RNAi-2 (31911R-3) reduce the ectopic TRE-dsRed in sal> ent2 by ~95% and ~90%, respectively. Note that the effect of scrib RNAi (5462R-2), AdoR RNAi (BL27536), ent2 RNAi-1 (GD7618) or egr RNAi-2 (GD45253) was not affected by concomitant expression of an RNAi against GFP (GFP-IR-1). This validates the use of co-expression as an assay for genetic interaction.

Flies were reared at a consistent density on standard cornmeal/agar media at 25 °C, unless indicated otherwise. MARCM (Mosaic analysis with a repressible cell marker) clones in wing imaginal discs were induced at the second instar (heat shock for 10 min at 37 °C 48–72 h after egg deposition (AED), dissection 48 h after clone induction). UAS-ent2 overexpression clones were generated using Actin-flp-out-Gal4. Clones were induced at the late second/early third instar (heat shock for 25 min at 37 °C 66–90 h AED, dissection 30 h after clone induction).

**Quantification of TRE-dsRed fluorescence**. Late third-instar larvae were staged by collecting them as white prepupae. Wing imaginal discs were fixed in 4% PFA for 20 min and washed in PBS. To quantify TRE-dsRed fluorescence intensity, we acquired confocal z-stacks (consisting of 5 optical slices taken at 5–10 μm intervals) through the whole sal-Gal4 domain (marked by expression of UAS-cd8GFP). TRE-dsRed fluorescence intensity within the sal-Gal4 domain of the collapsed z-stack images was quantified by digital image analysis using ImageJ. For each image, we subtracted the background intensity measured adjacent to the sal-Gal4 domain. A similar approach was used to quantify TRE-GFP fluorescence intensity in en-Gal4, UAS-nlacZ discs (Supplementary Fig. 1d). All P values were calculated using an unpaired two-tailed Student's t-test. Males and females were pooled for statistical analysis.

**Immunostaining**. White prepupae were used for wing disc immunostaining. Wing discs were fixed in 4% PFA, permeabilized with PBT and blocked in 2% NDS. Antibodies used in this study were chicken anti-GFP (1:1000; Abcam/ab13970), rat anti-Crumbs (1:500; generous gift of E. Knust[45]), mouse anti-Dlg (1:100, DSHB Hybridoma Product 4F3), mouse anti-LacZ (1:300; Sigma/G6282), rabbit anti-HA (1:500; Cell Signalling/#3724) and rabbit anti-Dcp-1 (1:200; Cell Signalling/#9578).

**RT-qPCR**. RNA was isolated from 10–15 male wing discs or from cultured HaCaT cells, using the RNeasy mini kit (Qiagen). For Figs. 2h, 4e, dissected wing discs were incubated in 1 mg/ml 2-chloroadenosine (CADO, Sigma) in Schneider medium (Gibco: 21720024) containing 10% heat-inactivated FBS with 10 ng/ml insulin (Sigma) for 4 h on a shaker prior to RNA preparation. cDNA was synthesized using the SuperScript III First-Strand Synthesis Supermix (Invitrogen). RT-qPCR was performed on the 7500 Fast Real-Time PCR System (Applied Biosystems) using iTaq Universal SYBR Green Supermix (Bio-Rad). RT-qPCR was performed in duplicate on each of at least four independent biological replicates. rp49 (for Drosophila) and GAPDH (for human cells) were used as normalization controls.

**RT-qPCR primer pairs**.
**Drosophila**
rp49-fwd 5′-CTTCATCCGCCACCAGTC-3′
rp49-rev 5′-CGACGCACTCTGTTGTCG-3′
puc-fwd 5′-GACGGCGACAGCGTGAGTC-3′
puc-rev 5′-GCCGTTGATGATGACGTCG-3′
dilp8-fwd 5′-CGACAGAAGGTCCATCGAGT-3′
dilp8-rev 5′-GATGCTTGTTGTGCGTTTTG-3′
TNF$^{egr}$-fwd 5′-GCATCCTCAGCCTCAAATGA-3′
TNF$^{egr}$-rev 5′-CCTGAAGCTCTGTGTGATTTCC-3′

*TNFR^grnd*-fwd  5-CCTTGAGCGGGCACAATCAC-3
*TNFR^grnd*-rev  5-TAACCGTTGTGGGCGTGGTA-3
**Human**
*GAPDH*  ordered from Sino Biological (Catalog: HP100003)
*PKG1*  ordered from Sino Biological (Catalog: HP100009)
*TNF-α*  ordered from Sino Biological (Catalog: HP100592)
*survivin*-fwd  5-AGTGAGGGAGGAAGAAGGCA-3
*survivin*-rev  5-ATTCACTGTGGAAGGCTCTGC-3

**Cell culture**. HaCaT cells (generous gift of B. Thompson) were cultured in DMEM (Gibco: 41966) containing 10% heat-inactivated FBS with 100 μg/ml streptomycin and 100 μg/ml penicillin. Cells were maintained in a 37 °C incubator at 5% atmospheric $CO_2$.

Adenosine, SCH58261, Caffeine and the JNK inhibitor SP600125 were obtained from Sigma. CGS15943 was from Santa Cruz Biotechnology and the PKA inhibitor cocktail was from Merck (#20-114). The reagents were used at the following concentrations: Adenosine 5 mM, SCH58261 80 μM, Caffeine 400 μM, CGS15943 400 μM, SP600125 40 μM and PKA inhibitor cocktail 0.5 μM. The presumed specificities of the adenosine receptor inhibitors are as follows: Adenosine, non-selective adenosine receptor agonist; SCH58261, adenosine receptor A2A antagonist; Caffeine, non-selective adenosine receptor antagonist and CGS15943, adenosine receptor A1 and A2A antagonist. Cells were pre-treated with SCH58261, Caffeine, CGS15943, SP600125 or PKA inhibitor cocktail for 1 h before other treatment. Cells were treated for a total of 12 h.

**Feeding and starvation experiments**. For feeding experiments, 2-day-old male flies (25–30 flies/vial) were kept in an empty vial containing a piece of 2.5 cm × 3.75 cm Whatman paper. 500 μl of 5% sucrose solution was used to wet the paper as feeding medium. To induce intestinal stress, 25 μg/ml Bleomycin (Merck Chemicals) was included in the feeding medium. Flies were transferred to fresh vials every day. Feeding experiments were performed for 14 days at 25 °C. For the dry starvation test, flies were placed in empty vials.

**Genotypes**. Figure 1a, e: *w1118*; *sal-Gal4, UAS-cd8GFP, TRE-dsRed(II)/+; UAS-GFP/+*. Figure 1b, f: *w1118*; *sal-Gal4, UAS-cd8GFP, TRE-dsRed(II)/+; UAS-GFP/UAS-scrib-RNAi*. Figure 1c, g: *w1118*; *sal-Gal4, UAS-cd8GFP, TRE-dsRed(II)/+; UAS-puc/UAS-scrib-RNAi*. Figure 1h: *w1118*; *sal-Gal4, UAS-cd8GFP, TRE-dsRed(II)/UAS-grnd-RNAi; UAS-scrib-RNAi/+*. Figure 1i: *w1118*; *sal-Gal4, UAS-cd8GFP, TRE-dsRed(II)/+; UAS-scrib-RNAi/UAS-egr-RNAi-2*. Figure 1j: *w1118*; *sal-Gal4, UAS-cd8GFP, TRE-dsRed(II), egr1/egr1; UAS-scrib-RNAi/+*. Figure 1l, o: *w1118*; *sal-Gal4, UAS-cd8GFP, TRE-dsRed(II)/+; UAS-scrib-RNAi/UAS-AdoR-RNAi*. Figure 1m, p: *w1118*; *sal-Gal4, UAS-cd8GFP, TRE-dsRed(II)/+; UAS-scrib-RNAi, AdoRKG/AdoR1*.

Figure 2a: *w1118*; *sal-Gal4, UAS-cd8GFP, TRE-dsRed(II)/+; UAS-Ent2/UAS-GFP*. Figure 2b: *UAS-bskDN/w1118 or Y; sal-Gal4, UAS-cd8GFP, TRE-dsRed(II)/+; UAS-Ent2/+*. Figure 2c: *w1118*; *sal-Gal4, UAS-cd8GFP, TRE-dsRed(II)/+; UAS-Ent2/UAS-AdoR-RNAi*. Figure 2e: *w1118*; *sal-Gal4, UAS-cd8GFP, TRE-dsRed(II)/UAS-AdoR*. Figure 2f: *w1118*; *sal-Gal4, UAS-cd8GFP, TRE-dsRed(II)/UAS-AdoR; UAS-Ent2/+*. Figure 2i: *w1118*; *sal-Gal4, UAS-cd8GFP, TRE-dsRed(II)/+; UAS-scrib-RNAi/+*. Figure 2j: *w1118*; *sal-Gal4, UAS-cd8GFP, TRE-dsRed(II)/UAS-AdoR; UAS-scrib-RNAi/+*.

Figure 3c: *w1118*; *sal-Gal4, UAS-cd8GFP, TRE-dsRed(II)/+; UAS-GFP/+*. Figure 3d: *w1118*; *sal-Gal4, UAS-cd8GFP, TRE-dsRed(II)/UAS-AdoR; UAS-GFP/+*. Figure 3e: *w1118*; *sal-Gal4, UAS-cd8GFP, TRE-dsRed(II), egr1/UAS-AdoR, egr1*. Figure 3f: *w1118*; *sal-Gal4, UAS-cd8GFP, TRE-dsRed(II)/UAS-AdoR; UAS-egr-RNAi-2/+*. Figure 3h: *w1118*; *sal-Gal4, UAS-cd8GFP, UAS-egr/+; TRE-dsRed(III)/UAS-GFP*. Figure 3i: *w1118*; *sal-Gal4, UAS-cd8GFP, UAS-egr/+; TRE-dsRed(III)/UAS-AdoR-RNAi*.

Figure 4b: *w1118*; *sal-Gal4/+; Eiger-GFP/+*. Figure 4c: *w1118*; *sal-Gal4/UAS-AdoR; Eiger-GFP/UAS-Ent2*. Figure 4d: *UAS-bskDN; sal-Gal4/UAS-AdoR; Eiger-GFP/UAS-Ent2*.

Supplementary Fig. 1e: *w1118*; *en-Gal4, UAS-nlacZ/TRE-GFP; UAS-scrib-RNAi/+*. Supplementary Fig. 1f: *w1118*; *en-Gal4, UAS-nlacZ/TRE-GFP; UAS-scrib-RNAi, AdoRKG/AdoR1*. Supplementary Fig. 1g: *w1118*; *TRE-dsRed(II)/TRE-dsRed(II)*. Supplementary Fig. 1h: *w1118*; *TRE-dsRed(II)/TRE-dsRed(II); FRT82B, scrib1/FRT82B, scrib1*. Supplementary Fig. 1i: *w1118*; *TRE-dsRed(II)/TRE-dsRed(II); FRT82B, scrib1, AdoR1/FRT82B, scrib1, AdoR1*. Supplementary Fig. 1l: *yw, hs-Flp, tub-Gal4, UAS-GFP; TRE-dsRed(II)/+; FRT82B, scrib1/FRT82B, tub-Gal80*. Supplementary Fig. 1m: *yw, hs-Flp, tub-Gal4, UAS-GFP; TRE-dsRed(II)/+; FRT82B, scrib1, AdoRKG/FRT82B, tub-Gal80*.

Supplementary Fig. 2b, c: *yw, hs-Flp; TRE-dsRed(II)/+; Act>y+>Gal4, UAS GFP/+*. Supplementary Fig. 2d, e: *yw, hs-Flp; TRE-dsRed(II)/+; Act>y+>Gal4, UAS GFP/UAS-Ent2*. Supplementary Fig. 2h: *w1118*; *sal-Gal4, UAS-cd8GFP, TRE-dsRed (II)/UAS-GFP-RNAi*. Supplementary Fig. 2i: *w1118*; *sal-Gal4, UAS-cd8GFP, TRE-dsRed(II)/UAS-GFP-RNAi; UAS-scrib-RNAi/+*. Supplementary Fig. 2j: *w1118*; *sal-Gal4, UAS-cd8GFP, TRE-dsRed(II)/+; UAS-scrib-RNAi/UAS-ent2-RNAi-1*. Supplementary Fig. 2l, m: *w1118*; *en-Gal4, UAS-nlacZ/+; Ent2-GFP, UAS-scrib-RNAi/TRE-dsRed(III)*. Supplementary Fig. 2n: *w1118*; *Ent2-GFP/Ent2-GFP*. Supplementary Fig. 2o: *w1118*; *en-Gal4, UAS-nlacZ/+; AdoR-GFP, UAS-scrib-RNAi/TRE-dsRed(III)*. Supplementary Fig. 2p, q: *w1118*; *en-Gal4, UAS-nlacZ/UAS-AdoR; UAS-Ent2/+*.

Supplementary Fig. 3a: *w1118*; *sal-Gal4, UAS-cd8GFP, TRE-dsRed(II)/+; UAS-GFP/UAS-scrib-RNAi*. Supplementary Fig. 3b: *w1118*; *sal-Gal4, UAS-cd8GFP, TRE-dsRed(II)/+; UAS-GαS-RNAi/UAS-scrib-RNAi*. Supplementary Fig. 3c: *w1118*; *sal-Gal4, UAS-cd8GFP, TRE-dsRed(II)/+; UAS-PKAmR*/UAS-scrib-RNAi*.

Supplementary Fig. 4a, b: *w1118*; *ap-Gal4/UAS-AdoR; Eiger-GFP/UAS-Ent2*.

## Data availability
All relevant data are available from the authors upon reasonable request.

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

## Acknowledgements

We thank A. Gould, G. Boulianne, D. Bohmann, P. Leopold, E. Knust and B. Thompson for sharing fly stocks and reagents. We also thank Gitta Stockinger, Venizelos Papayannopoulos and Andrew Bailey for comments on the manuscript. This work was supported by core funding (FC001204) from The Francis Crick Institute and fellowships from the Swiss National Science Foundation (number PBEZP3_142899) and EMBO (ALTF 20-2012) to I.P.

## Author contributions

I.P. designed the research, performed experiments and analyzed the data; I.P. and J.-P.V. wrote the manuscript.

## Additional information

**Competing interests:** The authors declare no competing interests.

