## [Peer Review File · Nature Communications]

Reviewers' comments:

Reviewer #1 (Remarks to the Author):

The ms of Vincent and colleagues identifies a role of Adenosine signaling in amplifying TNF/eiger production and, consequently, JNK signaling in polarity-impaired epithelial tissues and genetically dissect the epistatic relationship between TNF and Adenosine signaling. Authors initially define an experimental setup in which cell competition is being reduced and, by doing so, they are able to define the molecular elements mediating the role of Adenosine in amplifying TNF signaling. Authors, at the very end of the ms, use human cells and life span analysis to characterize whether the identified mechanism is conserved in human cells and whether it has an impact in tissue repair. The ms is nicely written, figures self-explanatory and the message is timely and interesting to the field and, as such, I believe this ms is a strong candidate for Nature Communication. I have one main comment that should be addressed by the authors. Authors propose in their model that Ado regulates Eiger levels upon epithelial disruption and use the Eiger-GFP to demonstrate that co-overexpression of both ent2 and AdoR is able to increase GFP levels. Data are validated with qPCRs. I wonder whether authors could analyze in scrib-depleted tissues whether Eiger is upregulated and whether this upregulation relies on Ado. The use of Eiger-GFP would be very visual.

Minor comments:

- (1) I do not understand the word "incidentally" in pg 4. Is it not expected that AdoR or TNFR removal has an impact in tumor growth, as this relies on JNK activity?
- (2) Authors might want to include the reference to Muzzopappa et al, 2012 where the Eiger-GFP line was carefully characterized in the wing primordium, and where a similar experimental set-up was used to avoid the "confounding" effects of cell competition.

Reviewer #2 (Remarks to the Author):

This manuscript by Poernbacher and Vincent shows that disruption of epithelial integrity in *Drosophila* wing disc causes activation of Adenosine receptor (AdoR) signaling by the extracellular adenosine, which is released from epithelial cells via the nucleotide transporter Ent. Elevated AdoR signaling transcriptionally upregulates TNF/Eiger and thereby causes JNK activation. They also show that TNF α mRNA is upregulated in human keratinocytes (HaCaT) treated with adenosine. Thus, the authors provide a new mediator of JNK signaling activation and a possible role of adenosine in chronic inflammation. These findings are interesting and novel. However, the data presented are too preliminary to support the authors' conclusion at this stage. Some crucial data presented need to be strengthened and there are several important experiments missing (detailed below). Particularly, the authors should carefully examine the correlation among JNK activity, AdoR signaling, and TNF expression.

Specific points:

1. Although the authors show that activation of AdoR induces eiger upregulation (Fig. 4), they do not investigate eiger mRNA level in polarity-deficient cells. This experiment is crucial for the author's conclusion that disrupted epithelial integrity causes TNF expression.
2. The authors claim that AdoR signaling induces eiger expression in polarity-deficient cells. If eiger mRNA is indeed upregulated by sal>scrib-RNAi, the authors should examine whether AdoR knockdown cancels eiger upregulation. Likewise, the authors should examine whether PKA signaling induces eiger expression in sal>scrib-RNAi or AdoR-overexpressing cells.
3. The authors claim that AdoR signaling is the inducer for eiger/TNF expression. However, the

data presented do not exclude the possibility that TNF expression is induced by JNK signaling activation. The authors should examine this possibility. Does JNK inhibition cancel TNF upregulation in *sal>scrib-RNAi* cells? Does Eiger- or Hep[act]-induced JNK activation cause TNF/eiger expression?

4. Relate to above, the authors show that Ent2+AdoR-overexpression causes JNK activation and TNF expression. The authors should examine whether or not TNF/eiger expression is mediated by JNK signaling in *sal>AdoR* cells (for example by coexpressing Bsk[DN] or Puc).

5. In supplementary Fig. 1b-i, the authors show that mutation of AdoR does not affect elimination of scrib mutant clones, while tumor size of scrib mutant tissue is reduced by AdoR mutation. Accordingly, the authors claim that AdoR signaling has a major role in JNK activation during chronic epithelial-polarity disruption. If this is the case, AdoR signaling must have a role in scrib mutant clones that evade from elimination (which can be tested for example by coexpressing p35). In addition, AdoR signaling must not be involved in JNK activation in a short-term induction of *sal>scrib-RNAi* (which can be tested for example by using temperature-sensitive Gal80). The authors should examine these to claim the conclusion.

6. The authors claim that disrupted epithelial integrity causes AdoR signaling activation. However, they do not provide evidence that AdoR or Ent2 overexpression does not affect apicobasal polarity. This should be examined to strengthen the conclusion.

7. The authors conclude that adenosine is released in the extracellular space via the nucleotide transporter Ent. Indeed, the data presented show that JNK signaling is activated in Ent-overexpressing compartment (*sal>gal4* in Fig. 2a-c) and clones (hsFLP-based mosaic clones in supplementary Fig. 2d and e). However, the authors do not provide data that Ent is involved in TNF expression and JNK activation in polarity-deficient cells. These should be examined. Does Ent knockdown reduce TNF expression and JNK activation in *sal>scrib-RNAi*? The authors should also examine whether Ent depletion suppresses wing defects caused by scrib knockdown (shown in Fig. 1f).

8. The authors show that adenosine is an inducer for TNF α expression in mammalian cells. In this condition, like flies, the authors should examine whether PKA is required for TNF expression downstream of AdoR. For example, does PKA inhibitor (e.g. H-89) cancel elevation of TNF mRNA in adenosine-treated HaCaT cells? Conversely, does 8-Bromo-cAMP (a cAMP analog) enhance TNF α expression?

9. In Fig. 1i and m, the data show that blocking AdoR expression suppresses JNK activation, although TRE-dsRed signal is still retained. Nevertheless, morphological defects in the adult wings are strongly suppressed by removing AdoR, just like UAS-Puc. The author should quantify the wing phenotype for proper interpretation of the data.

10. In supplementary fig. 3d, the authors show that knockdown of PKA-R subunit 1 (PKA-RNAi) has no effect on JNK activation in scrib-deficient cells. However, there are two regulatory subunits (R1 and R2) of PKA and they should examine the role of R2 subunit for JNK activation as well.

Reviewer #3 (Remarks to the Author):

Poernbacher and Vincent report that the response to the disruption of apical-basal polarity in the *Drosophila* larval epithelium includes signaling through the adenosine receptor to activate JNK signaling. The scribble loss of function model used here has been studied by many labs for a number of years. Therefore, it is refreshing to see a new signaling pathway in play, and how it connects to the much studied and very important JNK signaling; the link between AdoR signaling and JNK signaling will interest a wide audience. This study is potentially suitable for publication in

Nature Communications, in terms of its scope, impact and quality. I do have some concerns about the data should be addressed before proceeding further.

1. I commend the authors for assessing/validating the RNAi constructs, because not everyone does this important control. I do worry that when two RNAi constructs are used in the same cell, they could compete for the RNAi machinery and thereby reduce the efficacy of each. This is most concerning when the second RNAi construct is used to test for the rescue of the phenotype caused by the first (e.g. Fig 1k, n). The observed 'rescue' could be simply because the first RNAi construct is not as effective when the second one is present. For some of the experiments, the authors corroborate their double RNAi data with classical alleles (e.g. the last bar in Fig. 1 and in supplemental Fig. 1e that shows a partial rescue of *scrib*^{1/1} by *AdoR*^{1/1}, but unclear if statistically significant). But since some of the experiments are meant to address cell autonomous/autocrine effects, cell specific knock down by RNAi is the more relevant approach. The authors should confirm that RNAi is as effective in combination use as in single RNAi use. If not, adding UAS-dicer may help.

2. The title states that polarity-defective epithelial cells are the source of extracellular (e-) adenosine, yet Ade release is not directly assayed. It is implicated by the requirement for *ent1-3* in *sal* expressing cells but these are double RNAi experiments with possible alternate interpretation of the data (see #1). It remains possible that the source of adenosine is elsewhere. The minimal effect of AdeR overexpression in the absence of *scrib* knock down does suggest that the condition of *scrib* loss in some cells is needed for Ade signaling, but e-Ade could be coming from cells outside the *sal* domain.

3. Fig 4 shows that AdeR signaling can lead to *Egr* production, but in the context of experimentally stimulated AdeR signaling. It is important to show that this connection exists in the context of polarity disruption, which is the focus of this work. For example, is there a difference in *egr* expression in the *sal* domain (e.g. by in situ hybridization/reporter signal) in *sal*>*scrib* RNAi vs. *sal*>*scrib* RNAi, AdeR depleted discs?

Minor points:

Most of the analysis (Fig 1-4) uses a single GAL4 driver, *sal*-GAL4, to express RNAi. *sal*-GAL4 is active very late in wing disc growth/development. To confirm the generality of the findings, the authors should perform a couple of key experiments (e.g. Fig1n) using a different GAL4 driver that (1) is active earlier and (2) is active in a different part of the disc. *en*-GAL4 is already used in Fig S1 to show that *scrib* knock down does not alter the expression of Ade signaling components. Such discs could be analyzed for the requirement for Ade signaling.

Why such differences between males and females in some experiments (e.g. Fig. 2d, 3b, the first sets of bars) but not others (Fig. 3a, the first set of bars)? What was the reason for analyzing males and females separately in the first place?

In Fig S1, why isn't knocking down *AdoR* have a more severe effect on *puc* expression? If *Ado* signaling is really important to activate JNK, I would expect a stronger effect.

In Fig S2, why are *Crumbs*/*TRE*-dsRED signals different between A/P compartments in *h''* but similar in *f''*. It is also confusing to have both *Crumbs* and *TRE*-dsRED in the same colour. Please indicate the A/P boundary in *f-h''*.

In Fig. S4, please show a higher-magnification example of what the authors consider are macrophages, so we can see what is meant by 'identified on the basis of cell morphology'. It is impossible to see what the arrows are pointing to at this resolution/magnification.

RESPONSE TO REVIEWERS

Reviewers comments are reproduced verbatim in black. Our response is in blue.

Please note that the presentation of quantitative data (bar graphs) was adjusted in all the figures to comply with the requirements of Nature Communications.

Note also that the text changes in the manuscript are in blue font.

Reviewer #1 (Remarks to the Author):

The ms of Vincent and colleagues identifies a role of Adenosine signaling in amplifying TNF/eiger production and, consequently, JNK signaling in polarity-impaired epithelial tissues and genetically dissect the epistatic relationship between TNF and Adenosine signaling. Authors initially define an experimental setup in which cell competition is being reduced and, by doing so, they are able to define the molecular elements mediating the role of Adenosine in amplifying TNF signaling. Authors, at the very end of the ms, use human cells and life span analysis to characterize whether the identified mechanism is conserved in human cells and whether it has an impact in tissue repair. The ms is nicely written, figures self-explanatory and the message is timely and interesting to the field and, as such, I believe this ms is a strong candidate for Nature Communication. I have one main comment that should be addressed by the authors. Authors propose in their model that Ado regulates Eiger levels upon epithelial disruption and use the Eiger-GFP to demonstrate that co-overexpression of both *ent2* and *AdoR* is able to increase GFP levels. Data are validated with qPCRs. I wonder whether authors could analyze in *scrib*-depleted tissues whether Eiger is upregulated and whether this upregulation relies on Ado. The use of Eiger-GFP would be very visual.

As this (and the other reviewers) recognise, our model predicts that *eiger* transcription is expected to rise in *scrib*-deficient tissue. The existing *eiger* reporter is inserted at a location that is subject to position effect (Fig. 4b). This causes background expression around the center of the discs, which would mask the mild upregulation expected in *scrib* deficient cells (upregulation of JNK signalling is much weaker in discs expressing *scrib*-RNAi than those co-expressing *AdoR* & *Ent2*; Fig. 2f and 2i). For these reasons, we used qRT-PCR, which is more sensitive and quantitative, to measure gene expression. As shown in Fig. 4f, and as expected from our model, *eiger* mRNA was increased in wing imaginal discs obtained from *en>scrib*-RNAi larvae (relative to controls). This was brought back down to control levels in *en>scrib*-RNAi discs that are also *AdoR* mutant. The same was true in *en>scrib*-RNAi discs that concomitantly expressed *AdoR*-RNAi or *ent2*-RNAi. Co-overexpression of Puc had no such effect.

Minor comments:

(1) I do not understand the word “incidentally” in pg 4. Is it not expected that *AdoR* or TNFR removal has an impact in tumor growth, as this relies on JNK activity?

This word was used to indicate that tumour growth is of peripheral interest to our paper but we agree that it conveys the wrong impression and was replaced by “consistent with the established role of JNK signalling in tumour growth,”

(2) Authors might want to include the reference to Muzzopappa et al, 2012 where the Eiger-GFP line was carefully characterized in the wing primordium, and where a similar experimental set-up was used to avoid the “confounding” effects of cell competition.

Thank you for the suggestion.

Reviewer #2 (Remarks to the Author):

This manuscript by Poernbacher and Vincent shows that disruption of epithelial integrity in *Drosophila* wing disc causes activation of Adenosine receptor (AdoR) signaling by the extracellular adenosine, which is released from epithelial cells via the nucleotide transporter Ent. Elevated AdoR signaling transcriptionally upregulates TNF/Eiger and thereby causes JNK activation. They also show that TNF α mRNA is upregulated in human keratinocytes (HaCaT) treated with adenosine. Thus, the authors provide a new mediator of JNK signaling activation and a possible role of adenosine in chronic inflammation. These findings are interesting and novel. However, the data presented are too preliminary to support the authors' conclusion at this stage. Some crucial data presented need to be strengthened and there are several important experiments missing (detailed below). Particularly, the authors should carefully examine the correlation among JNK activity, AdoR signaling, and TNF expression.

Specific points:

1. Although the authors show that activation of AdoR induces eiger upregulation (Fig. 4), they do not investigate eiger mRNA level in polarity-deficient cells. This experiment is crucial for the author's conclusion that disrupted epithelial integrity causes TNF expression.

We addressed this point, also raised by reviewer 1, by measuring *eiger* mRNA levels with qRT-PCR. As shown in Fig. 4f, and as expected from our model, *eiger* mRNA increased in wing imaginal discs obtained from *en>scrib-RNAi* larvae.

2. The authors claim that AdoR signaling induces eiger expression in polarity-deficient cells. If eiger mRNA is indeed upregulated by *sal>scrib-RNAi*, the authors should examine whether AdoR knockdown cancels eiger upregulation.

Also shown in Fig. 4f, this was brought back down to control levels in *en>scrib-RNAi* discs that are also *AdoR* mutant. The same was true in

en>scrib-RNAi discs that concomitantly expressed *AdoR*-RNAi or *ent2*-RNAi. Co-overexpression of Puc had no such effect.

Likewise, the authors should examine whether PKA signaling induces *eiger* expression in *sal>scrib*-RNAi or *AdoR*-overexpressing cells.

We would indeed predict that PKA signalling be required for transcriptional upregulation of *eiger* in *scrib*-deficient discs. To test this prediction, we generated larvae co-expressing RNAi against *scrib* and a dominant negative PKA but, unfortunately, early lethality prevented us from obtaining third instar imaginal discs for expression analysis. A direct demonstration of PKA's requirement would have made a nice addition to the manuscript although we would argue that this is peripheral to our main finding, which is the involvement of the adenosine/*AdoR* in sensing epithelial disruption.

3. The authors claim that *AdoR* signaling is the inducer for *eiger*/TNF expression. However, the data presented do not exclude the possibility that TNF expression is induced by JNK signaling activation. The authors should examine this possibility. Does JNK inhibition cancel TNF upregulation in *sal>scrib*-RNAi cells? Does Eiger- or Hep[act]-induced JNK activation cause TNF/*eiger* expression?

This is an important point, which we had partially addressed in our initial submission. We showed then that overexpression of Puc, which strongly inhibits JNK signalling, did not prevent TNF^{eiger} upregulation in *sal>AdoR Ent2* discs (qRT-PCR experiment in old Fig. 4a, new Fig. 4a). In addition, we found that co-expression of JNK^{bskDN} did not prevent the upregulation of an Eiger-GFP in *sal> AdoR Ent2* discs (Fig. 4d, see point 4 below).

In response to the reviewer's suggestion, we have assayed TNF^{eiger} expression in *scrib*-RNAi-expressing discs that overexpressed Puc and found no significant difference with discs that solely express *scrib*-RNAi (Fig. 4f). This set of loss-of-function data strengthens our conclusion that JNK signalling itself does not contribute to TNF^{eiger} expression in polarity-deficient discs.

As suggested by the reviewer, we also assayed TNF^{eiger} expression in discs overexpressing Eiger. In this condition JNK signalling is strongly induced (Fig. 3h, j and panel a in the figure below) and yet no excessive TNF^{eiger} expression could be seen within the epithelium (panel b), confirming that JNK signalling on its own does not trigger TNF^{eiger} expression. The GFP signal seen in this condition stems from hemocytes that are attracted by cell death (see cell debris in higher magnification panel c) due to massive JNK activation. In light of the strength of the LOF data (and of the evidence presented in response to comment 4 below), we feel that these GOF results did not need to be included in the final manuscript. Confocal images in panels b and c are maximal intensity projections of z-stacks and thus show the signal from both the disc epithelium and attracted hemocytes on the surface of the disc.

TNF/Eiger overexpression (*sal>TNF^{Egr}* at 29°C) triggers JNK activation (panel a) but not TNF/Egr expression (panel b). Speckled green signal in panel b represents hemocytes (shown at high magnification in panel c) recruited to the area by the presence of apoptotic cells generated by excess JNK signalling. No excessive GFP signal is detected in the epithelium proper despite strong JNK signalling activation.

4. Related to above, the authors show that Ent2+AdoR-overexpression causes JNK activation and TNF expression. The authors should examine whether or not TNF/eiger expression is mediated by JNK signaling in *sal>AdoR* cells (for example by coexpressing Bsk[DN] or Puc).

The result of the experiment suggested by the reviewer was presented in our original submission (old Fig. 4a, new Fig. 4a). We showed then that overexpression of Puc did not prevent *TNF^{eiger}* upregulation (as assayed by qRT PCR) in discs overexpressing AdoR and Ent2 (*sal>AdoR ent2*). Because of the strong activation of *TNF^{eiger}* expression in response to co-expression of AdoR & Ent2, this effect was also seen with the Eiger-GFP reporter: Co-expression of JNK^{bskDN} did not prevent the upregulation of Eiger-GFP in *sal>AdoR Ent2* discs (Fig. 4d).

5. In supplementary Fig.1b-i, the authors show that mutation of AdoR does not affect elimination of scrib mutant clones, while tumor size of scrib mutant tissue is reduced by AdoR mutation. Accordingly, the authors claim that AdoR signaling has a major role in JNK activation during chronic epithelial-polarity disruption. If this is the case, AdoR signaling must have a role in scrib mutant

clones that evade from elimination (which can be tested for example by coexpressing p35).

The number of generations needed to perform this experiment would have added a substantial amount of time to the revision process (time was lost in obtaining a UAS-p35 insertion at the appropriate genomic location). In our opinion the result is not essential for the paper's message and one could argue that *scrib* mutant cells in an entirely mutant disc (Supplementary Fig. 1f-j) represents cells that evade elimination. For these reasons, we have not completed the suggested experiment.

In addition, AdoR signaling must not be involved in JNK activation in a short-term induction of *sal>scrib-RNAi* (which can be tested for example by using temperature-sensitive Gal80). The authors should examine these to claim the conclusion.

Transient gene inactivation by a combination of RNAi and Gal80ts is difficult to achieve because both processes take time to build up. As a result, we are unable to present results that meaningfully address the reviewer's suggestion.

6. The authors claim that disrupted epithelial integrity causes AdoR signaling activation. However, they do not provide evidence that AdoR or Ent2 overexpression does not affect apicobasal polarity. This should be examined to strengthen the conclusion.

Supplementary Fig. 2n, o shows that co-overexpression of AdoR and Ent2 has no detectable effect on apical-basal polarity (assayed by staining with anti-Dlg and anti-Crumbs).

7. The authors conclude that adenosine is released in the extracellular space via the nucleotide transporter Ent. Indeed, the data presented show that JNK signaling is activated in Ent-overexpressing compartment (*sal>gal4* in Fig. 2a-c) and clones (hsFLP-based mosaic clones in supplementary Fig. 2d and e). However, the authors do not provide data that Ent is involved in TNF expression and JNK activation in polarity-deficient cells. These should be examined. Does Ent knockdown reduce TNF expression and JNK activation in *sal>scrib-RNAi*? The authors should also examine whether Ent depletion suppresses wing defects caused by *scrib* knockdown (shown in Fig. 1f).

The role of Ent2 had already been partially addressed in our first submission where we showed that knockdown of equilibrative nucleoside transporters (*ent1, 2, 3*) or a hypomorphic *ent2* background blunted the activation of JNK signalling in *sal>scrib-RNAi* discs (old and new Fig2I). We have added an experiment showing that upregulation of *TNF^{Eiger}* in *scrib*-deficient discs (*en>scrib-RNAi*) is dampened by co-expression of *ent2-RNAi* (Fig. 4f). Consistently, the wing phenotype caused by *scrib-RNAi* was partially suppressed by *ent2-RNAi* (Supplementary Fig. 2f-i).

8. The authors show that adenosine is an inducer for TNF α expression in mammalian cells. In this condition, like flies, the authors should examine

whether PKA is required for TNF expression downstream of AdoR. For example, does PKA inhibitor (e.g. H-89) cancel elevation of TNF mRNA in adenosine-treated HaCaT cells? Conversely, does 8-Bromo-cAMP (a cAMP analog) enhance TNF α expression?

PKA activity was inhibited with a cocktail marketed by Merck (Catalog # 20-114). This led to significant reduction of *TNF* expression in adenosine-treated HaCaT cells (Fig. 4h). We also assessed the effect of PKA activation by 8-Br-cAMP. This caused a rise in *TNF* transcripts, albeit not as markedly as expected (figure below), perhaps because additional components activated by the AdoR contribute to the response. For the sake of completion, we also assessed the effect of a JNK inhibitor and found that it did not prevent upregulation of *TNF* expression in adenosine-treated HaCaT cells (Fig. 4h).

Treatment by 2-Br-cAMP (StemCell technologies, 1mM) leads to mild upregulation of TNF-alpha expression in HaCaT cells (Expression of PKG1, a house-keeping gene, is unchanged).

9. In Fig. 1i and m, the data show that blocking AdoR expression suppresses JNK activation, although TRE-dsRed signal is still retained. Nevertheless, morphological defects in the adult wings are strongly suppressed by removing AdoR, just like UAS-Puc. The author should quantify the wing phenotype for proper interpretation of the data.

Wing morphology is difficult to quantify. Since the presence of necrotic tissue correlates with reduced wing size, we measured the surface area of wings obtained from flies of the various genotypes. Using this crude assay, we confirmed that removal of *AdoR* suppresses the deleterious effect of polarity disruption (Supplementary Fig. 1b).

10. In supplementary fig. 3d, the authors show that knockdown of PKA-R subunit 1 (PKA-RNAi) has no effect on JNK activation in *scrib*-deficient cells. However, there are two regulatory subunits (R1 and R2) of PKA and they should examine the role of R2 subunit for JNK activation as well.

We found that RNAi against *PKA-R2* had no effect on JNK activation in *scrib*-deficient cells (Supplementary Fig. 3d).

Reviewer #3 (Remarks to the Author):

Poernbacher and Vincent report that the response to the disruption of apical-basal polarity in the *Drosophila* larval epithelium includes signaling through

the adenosine receptor to activate JNK signaling. The scribble loss of function model used here has been studied by many labs for a number of years. Therefore, it is refreshing to see a new signaling pathway in play, and how it connects to the much studied and very important JNK signaling; the link between AdeR signaling and JNK signaling will interest a wide audience. This study is potentially suitable for publication in Nature Communications, in terms of its scope, impact and quality. I do have some concerns about the data should be addressed before proceeding further.

1. I commend the authors for assessing/validating the RNAi constructs, because not everyone does this important control. I do worry that when two RNAi constructs are used in the same cell, they could compete for the RNAi machinery and thereby reduce the efficacy of each. This is most concerning when the second RNAi construct is used to test for the rescue of the phenotype caused by the first (e.g. Fig 1k, n). The observed 'rescue' could be simply because the first RNAi construct is not as effective when the second one is present. For some of the experiments, the authors corroborate their double RNAi data with classical alleles (e.g. the last bar in Fig. 1 and in supplemental Fig. 1e that shows a partial rescue of scrib1/1 by AdoR1/1, but unclear if statistically significant). But since some of the experiments are meant to address cell autonomous/autocrine effects, cell specific knock down by RNAi is the more relevant approach. The authors should confirm that RNAi is as effective in combination use as in single RNAi use. If not, adding UAS-dicer may help.

To address the reviewer's concern, we have combined key RNAi lines with a transgene expressing RNAi against GFP. This had no effect on the experimental results. This is illustrated in the figure below, prepared for the reviewer.

The effect of *scrib* RNAi (5462R-2), *AdoR* RNAi (BL27536), *ent2* RNAi-1 (GD7618) or *egr* RNAi-2 (GD45253) was not affected by concomitant expression of an RNAi against *GFP* (*GFP-IR-1*).

2. The title states that polarity-defective epithelial cells are the source of extracellular (e-) adenosine, yet Ade release is not directly assayed. It is implicated by the requirement for *ent1-3* in *sal* expressing cells but these are double RNAi experiments with possible alternate interpretation of the data (see #1 addressed/see above). It remains possible that the source of adenosine is elsewhere. The minimal effect of AdeR overexpression in the absence of *scrib* knock down does suggest that the condition of *scrib* loss in some cells is needed for Ade signaling, but e-Ade could be coming from cells outside the *sal* domain.

To demonstrate directly the release of adenosine is a tall order. Unfortunately, we have not been able to identify a suitable method to achieve this. However, we wish to highlight our observation (shown in Fig. 2i, j) that *AdoR* overexpression significantly potentiates the effect of *scrib*-RNAi. On its own, *AdoR* overexpression has only a minor effect (Fig. 2e), presumably for lack of extracellular adenosine. However, it markedly increases the effect of *scrib*-RNAi. This is most readily explained if *scrib* deficiency caused the release of adenosine that can engage with available *AdoR* at the cell surface. In addition, we wish to point out that polarity-deficient cells require the equilibrative adenosine transporter *Ent2* for full activation of JNK and TNF/*Eiger* expression, a strong indication that adenosine release takes place.

3. Fig 4 shows that AdeR signaling can lead to *Egr* production, but in the context of experimentally stimulated AdeR signaling. It is important to show that this connection exists in the context of polarity disruption, which is the focus of this work. For example, is there a difference in *egr* expression in the *sal* domain (e.g. by in situ hybridization/reporter signal) in *sal>scrib* RNAi vs. *sal>scrib* RNAi, AdeR depleted discs?

This experiment is also suggested by reviewer 1 and our response to him/her is copied below.

Our model predicts that *eiger* transcription is expected to rise in *scrib*-deficient tissue. The existing *eiger* reporter is inserted at a location that is subject to position effect (Fig. 4b, Supplementary Fig. 4a'). This causes background expression around the center of the discs, which would mask the mild upregulation expected in *scrib* deficient cells (upregulation of JNK signalling is much weaker in discs expressing *scrib*-RNAi than those co-expressing *AdoR* & *Ent2*; Fig 2 f and 2 i). For these reasons, we used qRT-PCR, which is more sensitive and quantitative, to measure gene expression. As shown in Fig. 4f, and as expected from our model, *eiger* mRNA was increased in wing imaginal discs obtained from *en>scrib*-RNAi larvae (relative to controls). This was brought back down to control levels in *en>scrib*-RNAi discs that are also *AdoR* mutant. The same was true in *en>scrib*-RNAi discs that concomitantly expressed *AdoR*-RNAi or *ent2*-RNAi. Co-overexpression of *Puc* had no such effect.

Minor points:

Most of the analysis (Fig 1-4) uses a single GAL4 driver, *sal-GAL4*, to express RNAi. *sal-GAL4* is active very late in wing disc growth/development. To confirm the generality of the findings, the authors should perform a couple of key experiments (e.g. Fig1n) using a different GAL4 driver that (1) is active earlier and (2) is active in a different part of the disc. *en-GAL4* is already used in Fig S1 to show that *scrib* knock down does not alter the expression of Ade signaling components. Such discs could be analyzed for the requirement for Ade signaling.

We have added several experiments based on the *en-Gal4* driver. See Supplementary Fig. 1c-e.

Why such differences between males and females in some experiments (e.g. Fig. 2d, 3b, the first sets of bars) but not others (Fig. 3a, the first set of bars)? What was the reason for analyzing males and females separately in the first place?

We noticed early on that males tend to have stronger phenotypes than females. The reason for the difference is unknown to us but we feel that it is important to tabulate the two sexes separately, as it prevents unnecessary data spread and provides directly comparable data sets. It might also spur other researchers to investigate the underlying biological basis of this phenomenon.

In Fig S1, why isn't knocking down *AdoR* have a more severe effect on *Puc* expression? If *Ado* signaling is really important to activate *JNK*, I would expect a stronger effect.

AdoR signalling is required for the full activation of *JNK*, but it is not the sole *JNK* activator during epithelial disruption. Indeed, the key message of our paper is that adenosine signalling boosts *JNK* signalling, by enhancing the production of *TNF*. This explains why loss of *AdoR* only causes partial suppression (as shown in Fig. 1n).

In Fig S2, why are *Crumbs*/*TRE-dsRED* signals different between A/P compartments in *h''* but similar in *f''*. It is also confusing to have both *Crumbs* and *TRE-dsRED* in the same colour. Please indicate the A/P boundary in *f-h''*.

In Fig S2, the A compartment (left) serves as a control for the P compartment (right), where gene activity is manipulated. Panels (f) and (h) from the old Fig S2 are from preparations stained separately and it appears that overall background is slightly higher in (*f''*) than in (*h''*). We have chosen a new prep (lower background) to replace the image shown in old Fig S2 (f). The results are now shown in Fig S2 (j, k). We have also separated the *Crumbs* and *TRE-dsRed* channels, as suggested. The A/P boundary is also indicated.

In Fig. S4, please show a higher-magnification example of what the authors consider are macrophages, so we can see what is meant by 'identified on the

basis of cell morphology'. It is impossible to see what the arrows are pointing to at this resolution/magnification.

A higher magnification panel has been included (Supplementary Fig. 4d).

REVIEWERS' COMMENTS:

Reviewer #1 (Remarks to the Author):

Reviewers have addressed my concerns in the revised version of the ms. I can now recommend publication of this ms in Nature Communications.

Reviewer #2 (Remarks to the Author):

In the revised manuscript by Poernbacher and Vincent, the authors have addressed most of the concerns I raised. However, there is one important issue that has not been addressed in the revised paper. The authors conclude in Fig. 4 that PKA induces TNF/eiger expression downstream of AdoR, but the data presented do not show that eiger is transcriptionally upregulated through PKA in the imaginal disc. During the revision, the authors tried to test whether eiger expression depends on PKA activity in *sal>scrib-RNAi* flies, but unfortunately larvae coexpressing *scrib-RNAi* and PKA[DN] showed early lethality. Thus, the authors claimed that it is technically difficult to obtain samples for eiger expression analysis. However, the authors present data that JNK activity (TRE-dsRed) can be reduced by PKA-C-RNAi expression in *sal>scrib-RNAi* flies, and they can indeed check eiger expression using those larvae. The authors would also be able to test whether expression of PKA catalytic domain causes eiger expression. Furthermore, the authors would be able to check if PKA[DN] or PKA-C-RNAi suppresses eiger expression caused by *sal>Ent2+AdoR* (shown in Fig. 4a-d). The conclusion that AdoR-PAK axis transcriptionally upregulates eiger has a great impact and thus the authors should adequately demonstrate this experimentally. Otherwise, the authors should omit or modify this part in the manuscript and the proposed model.

Reviewer #3 (Remarks to the Author):

In the revised version, Poernbacher and Vincent have addressed my concerns. I do have one clarification question based on the newly provided data, but, otherwise, this work is suitable for publication in Nature Communications. In the higher-magnification view provided in revised Fig. 4b, macrophages are described as having a 'distinct morphology'. By 'distinct morphology', do the authors mean smaller nuclei? If so, it should be stated, but more important, how can they tell smaller nuclei of macrophages apart from pyknotic nuclei of apoptotic disc cells?

RESPONSE TO REVIEWERS

Reviewers comments are reproduced verbatim in black. Our response is in blue.

REVIEWERS' COMMENTS:

Reviewer #1 (Remarks to the Author):

Reviewers have addressed my concerns in the revised version of the ms. I can now recommend publication of this ms in Nature Communications.

Reviewer #2 (Remarks to the Author):

In the revised manuscript by Poernbacher and Vincent, the authors have addressed most of the concerns I raised. However, there is one important issue that has not been addressed in the revised paper. The authors conclude in Fig. 4 that PKA induces TNF/eiger expression downstream of AdoR, but the data presented do not show that eiger is transcriptionally upregulated through PKA in the imaginal disc. During the revision, the authors tried to test whether eiger expression depends on PKA activity in *sal>scrib-RNAi* flies, but unfortunately larvae coexpressing *scrib-RNAi* and PKA[DN] showed early lethality. Thus, the authors claimed that it is technically difficult to obtain samples for eiger expression analysis. However, the authors present data that JNK activity (TRE-dsRed) can be reduced by PKA-C-RNAi expression in *sal>scrib-RNAi* flies, and they can indeed check eiger expression using those larvae. The authors would also be able to test whether expression of PKA catalytic domain causes eiger expression. Furthermore, the authors would be able to check if PKA[DN] or PKA-C-RNAi suppresses eiger expression caused by *sal>Ent2+AdoR* (shown in Fig. 4a-d). The conclusion that AdoR-PAK axis transcriptionally upregulates eiger has a great impact and thus the authors should adequately demonstrate this experimentally. Otherwise, the authors should omit or modify this part in the manuscript and the proposed model.

As stated in our earlier response to this reviewer, there is no suitable means of testing directly whether the PKA pathway is required for increased TNF/egr transcription in en> scrib Ri discs. This is because engrailed-gal4-driven PKAmR* or PKA-C-RNAi leads to early lethality. Weaker expression of these transgenes with *sal-Gal4* is viable but usage of this driver leads to another limitation, namely that *sal-Gal4*-driven *scrib-RNAi* does not trigger high enough TNF/Eiger expression for clear suppression to be detectable. With these limitations, we find ourselves unable to demonstrate directly the requirement for PKA. Nevertheless, all our other data (and existing literature on dAdoR) indicate that PKA is the most likely mediator of TNF/Eiger's transcriptional activation by adenosine. We added a comment summarising these points in the revised manuscript and removed PKA from the proposed model (Fig. 5).

Reviewer #3 (Remarks to the Author):

In the revised version, Poernbacher and Vincent have addressed my concerns. I do have one clarification question based on the newly provided data, but, otherwise, this work is suitable for publication in Nature Communications. In the higher-magnification view provided in revised Fig. 4b, macrophages are described as having a 'distinct morphology'. By 'distinct morphology', do the authors mean smaller nuclei? If so, it should be stated, but more important, how can they tell smaller nuclei of macrophages apart from pyknotic nuclei of apoptotic disc cells?

We added a clarifying comment to the Legend of Supplementary Fig. 4 as follows: 'The distinct morphology of macrophages (small round cells that are Egr-GFP positive)'.